# Oxidative cleavage and ammoxidation of organosulfur compounds via synergistic Co-Nx sites and Co nanoparticles catalysis

Huihui Luo[1,2,4], Shuainan Tian[1,3,4], Hongliang Liang[1], He Wang [3] ✉, Shuang Gao [1] & Wen Dai [1] ✉

The cleavage and functionalization of C−S bonds have become a rapidly growing field for the design or discovery of new transformations. However, it is usually difficult to achieve in a direct and selective fashion due to the intrinsic inertness and catalyst-poisonous character. Herein, for the first time, we report a novel and efficient protocol that enables direct oxidative cleavage and cyanation of organosulfur compounds by heterogeneous nonprecious-metal Co-N-C catalyst comprising graphene encapsulated Co nanoparticles and Co-Nx sites using oxygen as environmentally benign oxidant and ammonia as nitrogen source. A wide variety of thiols, sulfides, sulfoxides, sulfones, sulfonamides, and sulfonyl chlorides are viable in this reaction, enabling access to diverse nitriles under cyanide-free conditions. Moreover, modifying the reaction conditions also allows for the cleavage and amidation of organosulfur compounds to deliver amides. This protocol features excellent functional group tolerance, facile scalability, cost-effective and recyclable catalyst, and broad substrate scope. Characterization and mechanistic studies reveal that the remarkable effectiveness of the synergistic catalysis of Co nanoparticles and Co-Nx sites is crucial for achieving outstanding catalytic performance.

Organosulfur compounds are present ubiquitously in natural environment and fossil resources such as crude oil[1], coals[2], and natural products[3], and they also represent one of the most versatile building blocks in organic synthesis owing to their propensity to undergo a variety of transformations[4–6]. As such, enormous attention has been garnered to probe into the C-S bond activation modes[7] and mechanisms[8,9], and significant advances have been made in transition-metal-catalyzed cross-coupling and hydrodesulfurization reactions via C-S bond cleavage[10–13]. By contrast, the oxidative cleavage and transformation of C−S bonds in organosulfur compounds has still remained underexplored due to issues with chemoselectivity control and catalyst poisoning. Since the first reported oxidative cleavage of C−S bonds in sulfides to aldehydes using singlet oxygen by Corey in 1976[14], there have been continued attempts to allow chemoselectivity control of C-S

bond oxidative cleavage to carbonyl compounds[15–17]. However, current methods still remain limited in terms of scope and selectivity (Fig. 1a). Very recently, Lee et al. described an elegant photocatalysis strategy for controlled C−S bond oxidative cleavage of benzyl thiols to access aldehydes and ketones by a silver (II)-ligand complex[18] (Fig. 1b). Despite some progress, the oxidative cleavage and functionalization of C−S bonds are still in its infancy thus far, and there remains a high demand for exploring novel, controlled and widely applicable C-S bond oxidative cleavage strategies that enable the selective and diversified transformation of organosulfur compounds.

Nitriles are important structural motifs in the production of pharmaceuticals, agrochemicals, natural products, and functional materials[19,20]. Moreover, nitriles also sever as versatile intermediates for the synthesis of aldehydes, ketones, carboxylic acids, alcohols,

[1]Dalian Institute of Chemical Physics, Chinese Academy of Sciences, Dalian, PR China. [2]University of Chinese Academy of Sciences, Beijing, PR China. [3]School of Chemistry and Materials Science, Liaoning Shihua University, Fushun, PR China. [4]These authors contributed equally: Huihui Luo, Shuainan Tian. ✉e-mail: hewang@lnpu.edu.cn; daiwen@dicp.ac.cn

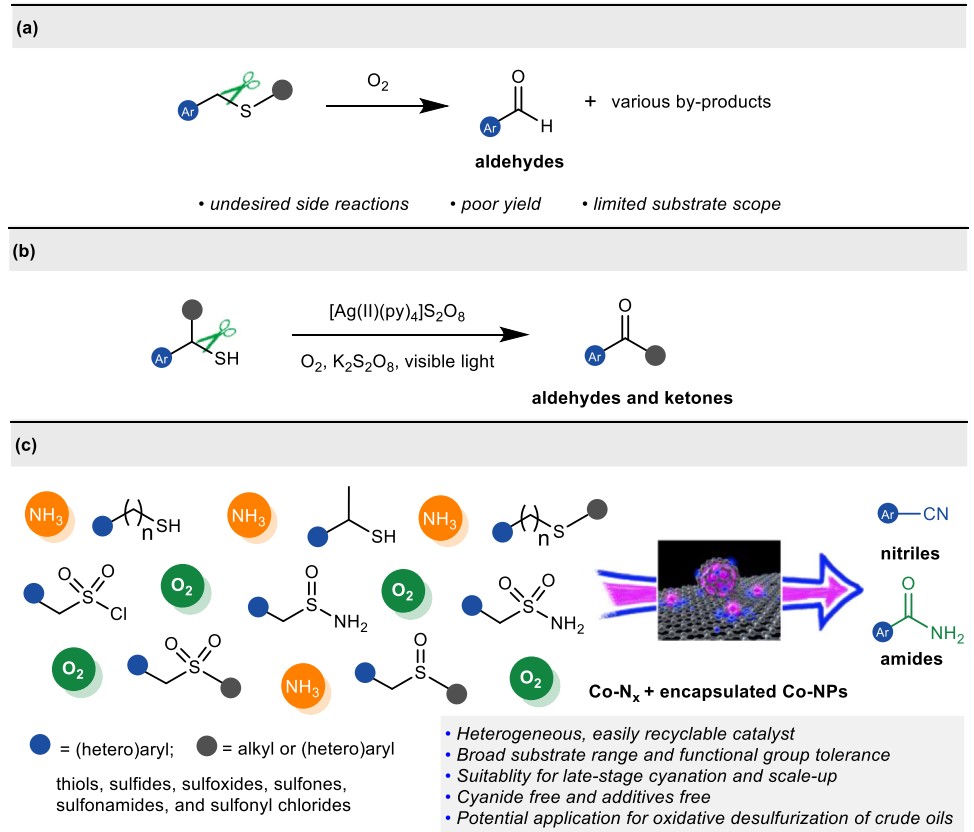

**Fig. 1 | Approaches to oxidative cleavage and functionalization of C-S bonds. a** Uncontrolled C-S bond oxidative cleavage. **b** Controlled oxidative cleavage of thiols to aldehydes and ketones. **c** This work: controlled oxidative cleavage and ammoxidation of organosulfur compounds.

amides, and heterocycles[21]. Traditionally, general strategies for the preparation of nitriles include Sandmeyer reaction[22–24], transition-metal-catalyzed cyanation of aryl cyanation under homogeneous conditions, complications associated with stability, separation and recyclability of catalyst, large-scale production, and metallic impurities are still a challenge halides[25,26], and direct cyanation of C–H bonds[27–29]. While the aforementioned methods are successful in achieving arene. Moreover, toxic organic or inorganic cyanides and stoichiometric metal oxidant were typically used, thus posing a significant safety and environment concern. Therefore, a sustainable heterogeneous and one-step direct construction of aryl nitriles under cyanide-free conditions remains highly desirable.

In the last decades, the development of non-noble metal-based heterogenous catalysts for organic synthesis has become a prime topic and is crucial for the advancement of green and sustainable industrial process[30–39]. Among them, cobalt-based nanoparticles or single atom catalysts (SACs) are much more attractive due to the earth-abundant, non-toxic, biocompatible, and environmentally benign characteristics of Co[35–39]. Recently, we reported the first heterogeneous catalytic oxidative cleavage and esterification of C-C bonds in alcohols using $O_2$ as terminal oxidant by cobalt/N-doped carbon catalyst[40]. Driven by some specific cobalt/N-doped carbon catalysts capable of oxidatively cleaving the strong C-C bonds, and as a continuation of our long-time interest in value-added transformation of chemical feedstocks[41–47], we intended to investigate the utilization of the heterogeneous cobalt-based catalyst for the direct conversion of abundant sulfur-containing feedstocks, and hoped to accomplish one of the only catalytic example of oxidative cleavage and cyanation of C-S bonds under heterogeneous conditions. Herein, for the first time, we report a practical and sustainable strategy for nitrile formation by heterogeneous Co-NC catalysts comprising graphene encapsulated Co nanoparticles and Co-Nx sites on N-doped porous carbon via cleavage and cyanation of C-S

bonds in organosulfur compounds with oxygen as environmentally benign oxidant and ammonia as nitrogen source. A wide variety of thiols, sulfides, sulfoxides, sulfones, sulfinamides, and sulfonamides are viable in this reaction, enabling access to diverse nitriles under cyanide-free conditions. Noteworthy is that modifying the reaction conditions also enables the cleavage and amidation of organosulfur compounds, allowing access to structurally diverse amides. Further studies reveal that the high efficiency of this catalytic system mainly originates in synergistic catalysis of Co nanoparticles and Co-Nx sites. Moreover, the easily prepared Co-NC catalysts could be recovered and reused at least five times without loss of efficiency. Therefore, the present method not only expands the range of application of the sulfur-containing feedstocks, but also provides a new strategy for the divergent, late-stage functionalization of complex molecules (Fig. 1c).

## Results

### Catalyst synthesis

The Co−NC catalysts were prepared by a support-sacrificial approach according to our previous report[40]. Briefly, a Co(phen)x (phen = 1,10-phenanthroline) complex precursor was first supported on silica and then the mixture was submitted to pyrolysis at 700−900 °C in $N_2$ for 2 h, after which the silica support was removed by treating the material with hydrofluoric acid. The samples pyrolyzed at different temperatures are denoted as Co−NC−X (X = pyrolysis temperature).

### Catalyst activity tests

Initially, phenylmethanethiol **s1** was chosen as the model substrate for oxidative cleavage and cyanation of C-S bonds, and the reaction was performed at 150 °C with $O_2$ and aq. $NH_3$ as the oxidant and nitrogen source, respectively. As shown in Table 1, in the absence of the catalyst, no desired nitrile **1** was observed, instead producing disulfide **a** as the major product (entry 1). Among the various Co-NC-X catalysts tested

## Table 1 | Optimization of the reaction conditions[a]

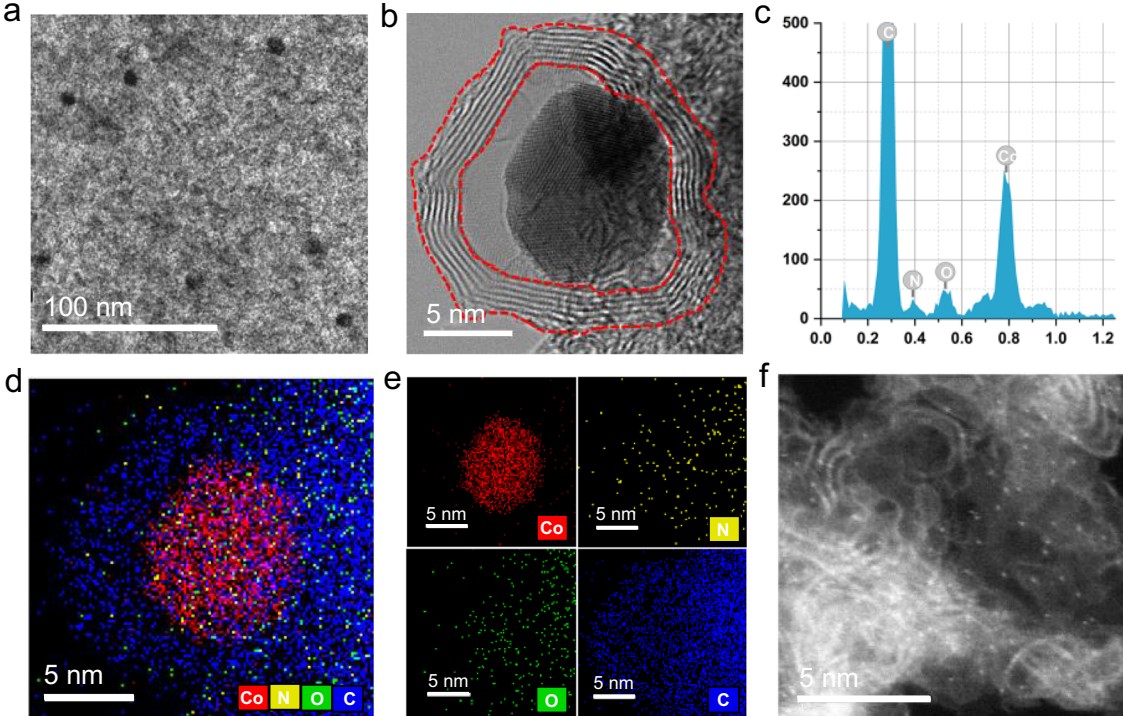

| Entry | Catalysts | Conv. (%)[b] | Yield (%)[b] | |
|---|---|---|---|---|
| | | | a | 1 |
| 1 | — | >99 | 89 | — |
| 2 | Co-NC-700 | >99 | 20 | 74 |
| 3 | Co-NC-800 | >99 | — | 92 |
| 4 | Co-NC-900 | >99 | — | 94 |
| 5 | Co(OAc)$_2$·4H$_2$O | >99 | 72 | — |
| 6 | 1,10-phenanthroline | >99 | 76 | 5 |
| 7 | Co(phen)x | >99 | 70 | 7 |
| 8 | Co(phen)x@SiO$_2$−900 | >99 | 66 | 23 |
| 9[c] | Co-NC-900 | >99 | 6 | 88 |
| 10[d] | Co-NC-900 | >99 | 3 | 87 |
| 11[e] | Co-NC-900 | >99 | 58 | 10 |
| 12[f] | Co-NC-900 | >99 | — | 87 |

[a]Reaction conditions: phenylmethanethiol (0.25 mmol), catalyst (5.5 mol%), 25-28 wt% aq. NH$_3$ (155 μL), t-amyl alcohol (2 mL), 1.0 MPa O$_2$, 150 °C, 6 h.
[b]Determined by GC analysis using biphenyl as internal standard and the products were confirmed by GC-MS.
[c]130 °C.
[d]4 mol% catalyst.
[e]1.0 MPa Ar.
[f]1.0 MPa air.

**Fig. 2 | Characterization of the Co-NC-900 catalyst. a** TEM image. **b** HAADF-STEM image of cobalt/graphene core-shell structure in Co-NC-900. **c** EDX spectrum. **d**, **e** overlapped EDX map followed by separate map images. **f** HAADF-STEM image of Co-NC-900 in the nanoparticle-free region, in which single Co atoms are clearly seen.

(entries 2–4), Co-NC-900 gave the best yield (94% yield, entry 4), suggesting the catalyst activity is significantly influenced by pyrolysis temperature. Several control experiments were also performed with Co(OAc)$_2$·4H$_2$O, 1,10-phenanthroline or Co (II) complex of 1,10-

phenanthroline (Co(phen)x) as catalyst (entries 5-7). Likewise, they all exhibit poor activity toward cyanation reaction. It was noted that the Co(phen)x@SiO$_2$−900 exhibited inferior reactivity than Co−NC−X (entry 8), indicating that the removal of template is in favor of

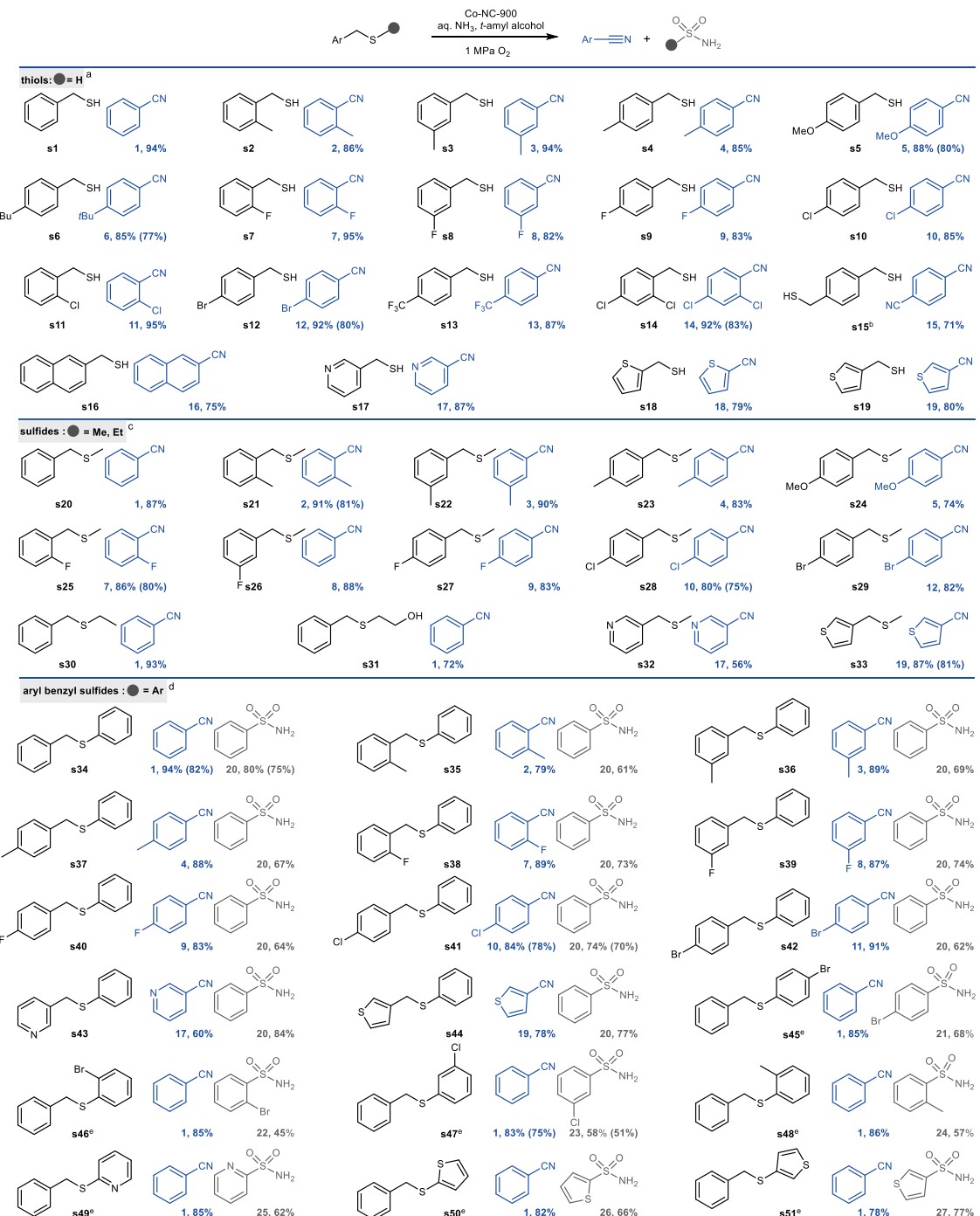

**Fig. 3 | Substrate scope of thiols as well as sulfides and benzyl sulfides.** Reaction conditions: substrates (0.25 mmol), *t*-amyl alcohol (2 mL), 1.0 MPa O₂, Yield was determined by GC analysis using biphenyl as internal standard and the product was confirmed by GC-MS, and isolated yields are given in parentheses. [a]Co-NC-900 (5.5 mol%), 25-28 wt% aq. NH₃ (155 μL). 150 °C, 6 h. [b]Co-NC-900 (11 mol%), 25-28 wt% aq. NH₃ (233 μL), acetonitrile (2 mL), 150 °C, 6 h. [c]Co-NC-900 (11 mol%),150 °C, 24 h. [d]Co-NC-900 (11 mol%), 25-28 wt% aq. NH₃ (233 μL), 150 °C, 24 h. [e]Co-NC-900 (11 mol %), 25-28 wt% aq. NH₃ (233 μL), 160 °C, 48 h.

increasing specific surface area, endowing the catalyst to expose more active sites. A decrease of either catalyst loading or reaction temperature led to a lower yield of **1** (entries 9 and 10). Only 10% yield of **1** was furnished when the reaction was conducted under argon atmosphere, demonstrating that O₂ is indispensable in this transformation (entry 11). Interestingly, 87% yield of **1** can also be achieved when the oxygen was replaced with air, highlighting the practical utility of this catalyst system (entry 12). For further screening of the conditions, please see Supplementary Tables 1, 2, and we finally adopted the

conditions of entry 1 (155 μL aq. NH₃, 2 mL *t*-amyl alcohol, 1.0 MPa O₂, 150 °C, 6 h) as standard for the ensuing study of scope.

## Catalyst characterization

Given such impressive findings, we next investigated the structural properties of the catalyst Co-NC-900 by means of comprehensive technical skills. Transmission electron microscopy (TEM) analysis revealed the formation of cobalt particles with an average size of 10 nm (Fig. 2a). The high-angle annular dark-field scanning transmission

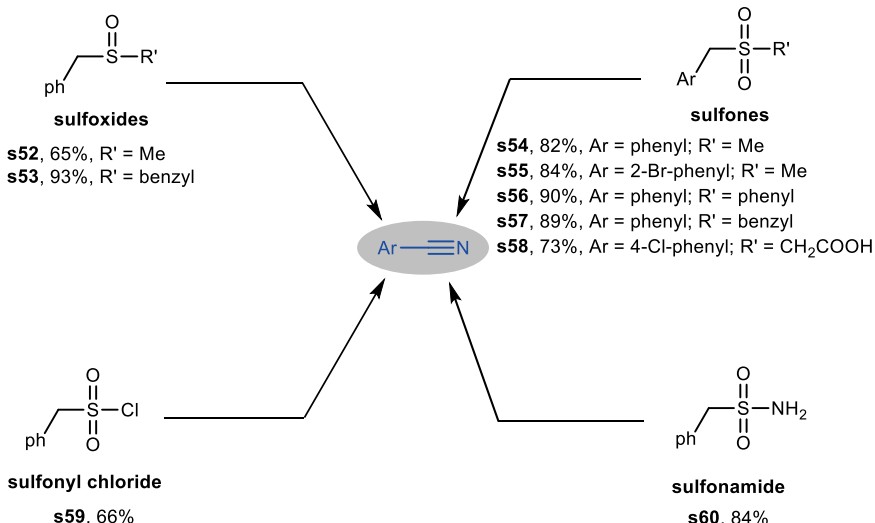

**Fig. 4 | Substrate scope of other types of organosulfur compounds.** Reaction conditions: substrates (0.25 mmol), Co-NC-900 (11 mol%), 25–28 wt% aq. NH$_3$ (155 µL), *t*-amyl alcohol (2 mL), 1.0 MPa O$_2$, 150 °C, 24 h. Yield was determined by GC analysis using biphenyl as internal standard and the product was confirmed by GC-MS.

electron microscopy (HAADF-STEM) images showed that the particles were protected by graphene-like shells, comprising typically 6–10 layers with a thickness of 2.1–3.5 nm (Fig. 2b). The composition of the nanostructures was further verified by energy dispersive X-ray (EDX) spectroscopy, with N, O, and Co detected (Fig. 2c). Nanoscale element mapping (Fig. 2d, e) demonstrated that cobalt was concentrated in the core region, whereas C, N, and O were distributed throughout the entire particle, confirming the core-shell structure. Presumably, carbonization of the nitrogen-containing ligand leads to the formation of nitrogen doped graphene-like shells. In addition, we also found numerously bright dots distributed on the carbon matrix in the nanoparticle-free region, indicating the existence of single Co atoms (Fig. 2f). The details of structural characterization of Co-NC-900 with SEM, BET, XPS, and XAS can be found in our previous reports[40]. Taking all characterization results into account, we can conclude that the as-prepared catalyst Co-NC-900 comprises core–shell structured nanoparticles with metallic Co as the core and layers of graphitic carbon as the shell and atomically dispersed Co-Nx sites as well.

## Substrate scope

With the optimal reaction conditions identified, we then explored the reaction scope with respect to thiols. As shown in Fig. 3, a variety of benzyl thiols smoothly underwent oxidative cleavage to afford the desired nitriles in excellent yields (82–95%), regardless of the electronic nature of the substituents on the aromatic ring (**s1-13**). Ortho-substituted benzyl thiols exhibited well-matched reactivity to meta- and para- isomers, indicating steric hindrance has no significant effect on the reaction efficiency (**s2-4, s7-9**, and **s10-11**). The reaction with the benzyl thiol bearing two substituents on the aromatic rings also proceeded well to give the corresponding product in excellent yield (**s14, s15**). Additionally, aryl framework of benzyl thiol can be extended to a naphthalene-derived system to afford the desired product in good yield (**s16**). Notably, thiols with nitrogen- and sulfur-containing heterocycles such as pyridine and thiophene are also well tolerated and successfully cleaved to the corresponding heteroaromatic nitriles in excellent yields, demonstrating the potential for this protocol to be used in the synthesis of bioactive compounds (**s17-19**).

Next, the reaction scope for the oxidative cleavage and cyanation of sulfides was also investigated. As shown in Scheme 2, the benzyl methyl sulfide **s20** and its derivatives were all compatible, generating products in good to excellent yields, regardless of the electronic properties and position of the substituents on the aromatic ring (**s21-**

**29**). Excellent yields were preserved even when a methyl group on the sulfur atom was replaced by longer alkyl or functionalized alky (**s30, s31**). Heteroaromatic sulfides, including pyridine-, furan-, and thiophene-substituted sulfides were also found to be suitable substrates, thus affording the desired nitriles in satisfactory yields (**s32, s33**).

Encouraged by the above results, we then turned our attention to sterically hindered sulfide to further examine the generality of our catalytic system. As shown in Fig. 3, different kinds of aryl benzyl sulfides were found to undergo efficient cleavage reaction. The reactions were suitable with an electronically and sterically diverse set of substituents, affording the targeted nitriles in good to excellent yields (79–91%), along with benzenesulfonamide **20** in good yields (61–74%) (**s34-42**). Moreover, sterically encumbered sulfides containing heterocycles, including pyridine and thiophene, were smoothly transformed to the corresponding nitriles and sulfonamides with excellent selectivity (**s43, s44**). It is well known sulfonamides are valuable structural motifs in medicinal and agrochemical agents due to their chemical and metabolic stability, carboxyl bioisosterism, and high level of biological activity[48]. The classical approach to prepare sulfonamides involves the reaction between amine nucleophiles and sulfonyl chlorides[49]. However, sulfonyl chlorides are not widely available and are toxic, unstable reagents. Thus, in addition to cyanation of sulfides, the current catalytic system also offered a valuable alternative to direct synthesis of sulfonamides from the sulfides and amines, two readily available and inexpensive commodity chemicals, to prepare sulfonamides, further highlighting the broader adaptability of this protocol (**s45-s51**).

Besides thiols and sulfides, we anticipated that our protocol might also be applicable to other types of organosulfur compounds. A series of sulfoxides, sulfones, sulfonamides, and sulfonyl chlorides were examined. As shown in Fig. 4, both benzyl alkyl and aryl benzyl sulfoxides smoothly underwent cleavage to give the corresponding benzonitrile (**s52, s53**). Subsequently, a series of structurally diverse of sulfones were subjected to this protocol, delivering the desired nitriles in good yield (**s54-s58**). Moreover, sulfonyl chloride and sulfonamides also proved to be viable substrates, affording the desired products in good yields (**s59, s60**).

Amides, as another important class of compounds, are widely applied in organic chemistry, materials science, polymers, agrochemical, and pharmaceutical industry[50–52]. As such, development of efficient methodologies towards amides continues to be scientifically

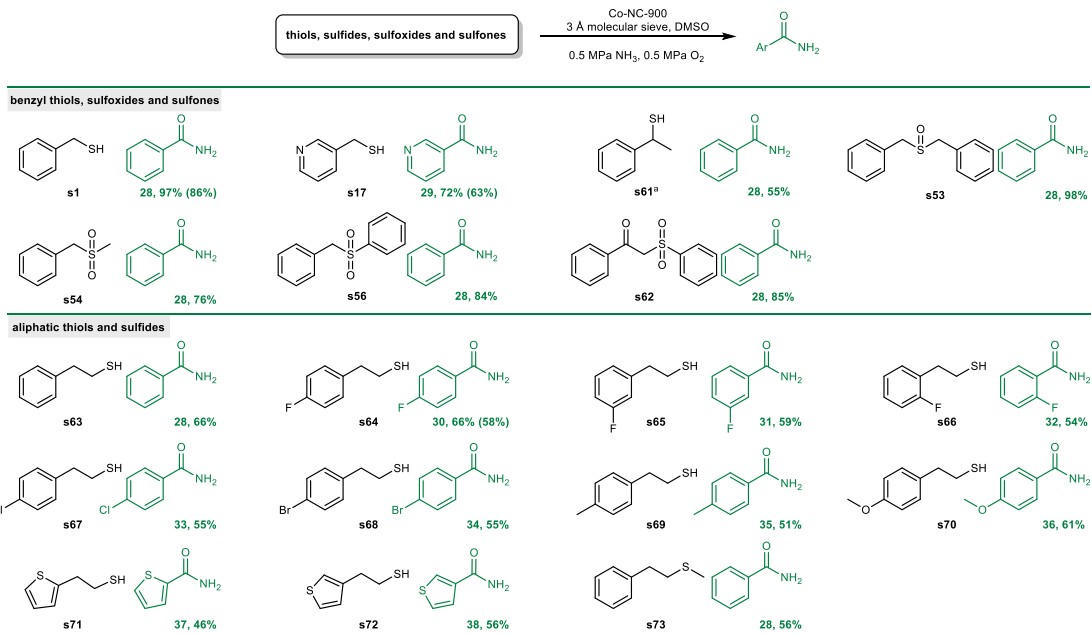

**Fig. 5 | Substrate scope for the synthesis of amides.** Reaction conditions: substrate (0.25 mmol), Co-NC-900 (5.5 mol%), 3 Å molecular sieve (100 mg), DMSO (1 mL), 0.5 MPa NH$_3$, 0.5 MPa O$_2$. The detailed experimental procedures are provided in the Supporting Information. Yield was determined by GC analysis using biphenyl as internal standard and the product was confirmed by GC-MS. [a]Co-NC-900 (11 mol%), 25-28 wt% aq. NH$_3$ (155 μL), 170 °C, 48 h.

interesting and attracts a broad attention of the synthetic community. In this context, we wondered whether the current methodology could also be extended to oxidative cleavage and amidation of organosulfur compounds. The results are summarized in Fig. 5. A significant increase in selectivity for amides formation can be achieved through tuning the reaction conditions. A variety of benzyl thiols (**s1, s17, s61**), sulfoxides (**s53**), and sulfones (**s54, s56, s62**) are suitable substrates to deliver the corresponding amides in synthetic useful yields. Compared to benzyl thiols and sulfides, the cleavage and amidation of unactivated aliphatic thiols and sulfides is undoubtably more challenging. Delightedly, the current catalytic system is well suited for amidation of aliphatic sulfur-containing compounds. Both phenyl- and thiophene-substituted ethanethiols (**s63-72**) can be efficiently cleaved to provide the corresponding amides in good yields. Furthermore, the aliphatic sulfide (**s73**) also proved to be viable substrate.

### Hot filtration experiment and recycling experiments

To gain a deeper understanding of whether or not the active species is heterogeneous in nature, a hot filtration experiment was conducted during the reaction process. After 1.5 h of oxidative cleavage reaction of phenylmethanethiol **s1**, the catalyst (Co-NC-900) was quickly filtered, and the reaction solution was allowed to stir for an additional 4.5 h. It was found that after filtration of the catalyst, the reaction seems to stop, and the yield remains the same as after 1.5 h of reaction, confirming the heterogeneous role of the catalyst during the reaction (Supplementary Fig. 1). Moreover, the reaction of **s1** was performed under standard conditions to demonstrate the reusability of Co-NC-900 catalyst. After the completion of the reaction, the catalyst could be easily recovered by centrifugation and reactivated by pyrolyzing at 900 °C for 2 h, and was subsequently applied to successive six cycles without loss of catalytic efficiency (Fig. 6a). Furthermore, we investigated the composition and structure of the used catalyst by TEM (HRTEM) (Supplementary Fig. 2) and XRD (Supplementary Fig. 3), disclosing that no discernable changes are observed compared with the fresh catalyst. The results unambiguously demonstrate the catalyst possess excellent stability.

### Additional applications

To further demonstrate the practicality of the developed method, a gram-scale reaction was performed. When the reaction of thiol **s1** was scale up to 1.01 g (8 mmol), the desired product **1** was formed in 87% yield even though the catalyst loading was reduced (Fig. 6b). Additionally, oxidative cleavage of sulfide **s33** was also performed on a gram scale, giving **1** in 93% yield, along with benzenesulfonamide **19** in 65% yield (Fig. 6b). Next, the newly developed transformation was proved to be different from traditional organic reactions, in which pure starting materials are needed. When the mixture of benzyl thiols, benzyl sulfide, and benzyl phenyl sulfide were subjected to the current protocol, the nitrile was obtained in 88% yield, which emphasizes the potential application in preparing a single nitrile product from a crude mixture of organosulfur compounds (Fig. 6c). Air is cheaper, safer, and more easily handled in comparison to pure oxygen. Then, we performed the reactions of oxidative cleavage and cyanation of different types of organosulfur compounds using air as oxidant. As depicted in Fig. 6d, a wide variety of thiols (**s1, s3, s11, s17**), sulfides (**s20, s33, s34**), sulfoxides (**s53**), sulfones (**s56**), and sulfonamides (**s60**) can be efficiently transformed into the corresponding nitriles in high to excellent yields under an air atmosphere. A general platform that enables late-stage functionalization of complex molecules in a single reaction vessel would be particularly powerful in drug discovery programs. Having validated the current methodology on simple organosulfur compounds, we anticipated that our protocol might be applicable to late-stage cyanation. To this end, the thiol-containing tonalid derivative **s74** and (+)-δ-tocopherol derivative **s75** were conducted under standard rection conditions, affording the desired product **39** and **40** in 59% and 64% isolated yield, respectively, highlighting the potential impact of our protocol on generating complex nitrile architectures to accelerate lead compound discovery (Fig. 6e). The desulfurization of heavy oils is one of the challenges faced during refining to produce transportation fuels and petrochemicals. In order to further explore the application potential of this protocol, attempts were made to accomplish the oxidative transformation of inert aliphatic organic sulfur compounds such as aliphatic thiols (**s76-s79**), disulfides (**s80**) and sulfides (**s81**) which represent the main source of sulfur found in

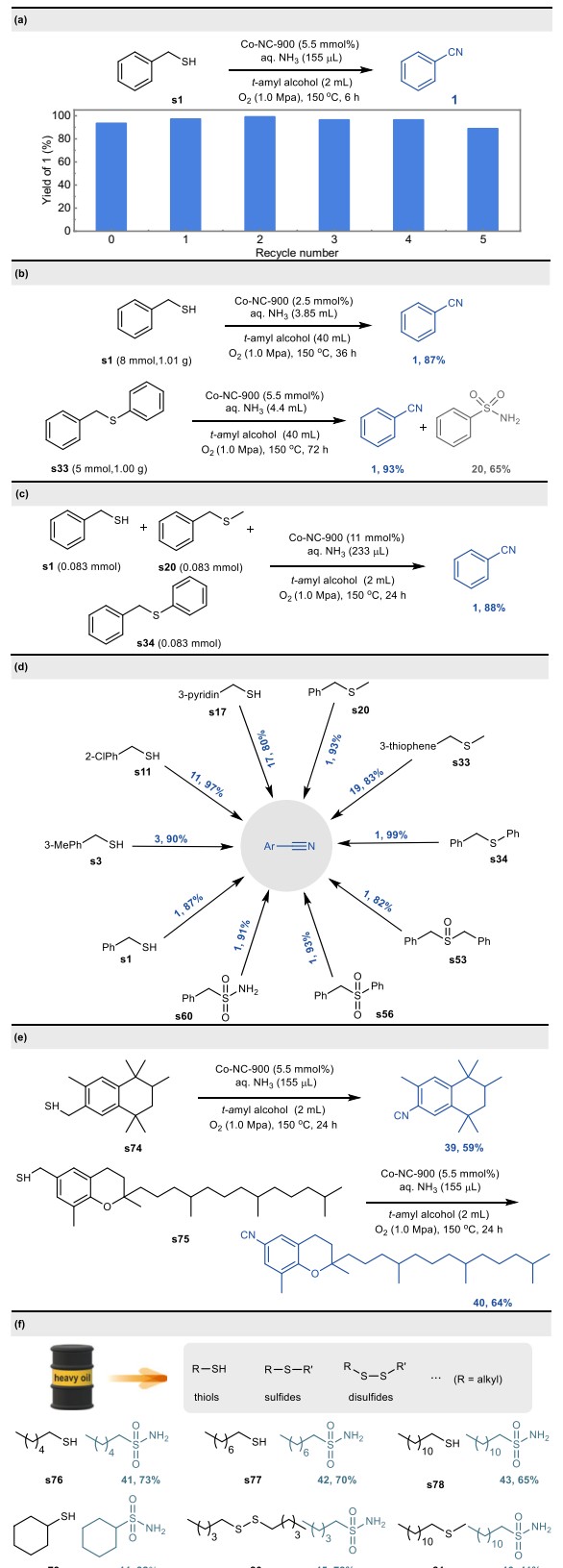

**Fig. 6 | Recycling experiments and additional applications. a** Recycling experiments. **b** Gram-scale reaction. **c** Cleavage and cyanation of mixed organosulfur compounds in a one-pot reaction. **d** Cyanation of organosulfur compounds using air as oxidant. **e** The late-stage modification of complex bioactive molecules. **f** Oxidative desulfurization of sulfur-containing compounds in heavy oils.

heavy oils (Fig. 6f). Although low yield (<10%) of target nitriles or amides was obtained for the oxidative cleavage of C-S bond in these inert aliphatic organosulfur compounds, high conversion could be achieved to furnish the high-value-added sulfonamides in good yield. Sulfonamides are more polar than other hydrocarbons in crude oils, and thus can be simplify removed from oil phase by extractive desulfurization. These results demonstrate that the current catalytic oxidation system can provide an alternative strategy for the oxidative desulfurization of heavy oil.

## Reaction mechanism

In order to gain insights into the mechanism of the cleavage and cyanation reaction of benzyl thiols, the time course for the conversion of phenylmethanethiol **s1** was performed under standard conditions (Fig. 7a). It was found that **s1** was fully consumed within 0.5 h. Disulfide **a** was observed but was further oxidized after the peak at 0.5 h. Aldehyde **b** has been detected and reaches its summit at around 1 h. Both **a** and **b** were completely converted to the desired final product **1** at 6 h with the concurrent generation of small amount of benzamide **28**. To distinguishing these compounds, formed during the reaction process, being reaction intermediates or byproducts, a set of control experiments were conducted (Fig. 7b). Under the standard reaction conditions, both disulfide **a** and aldehyde **b** were transformed into the target nitrile product **1** in excellent yields. Thus, we conclude that disulfide and aldehyde are clearly the intermediates of this transformation. It is well known that dehydrative condensation of aldehyde and ammonia would result in an aldimine via a hemiaminal intermediate, followed by oxidative dehydrogenation of the aldimine to afford nitrile[53–55]. To unveil the possible pathways for amide formation, the benzonitrile **1** was subjected to the standard reaction conditions. Interestingly, no benzamide **28** was observed, thereby ruling out the nitrile hydration process. Therefore, the present amidation proceeds through oxidative cleavage of thiol to hemiaminal, followed by the direct oxidative dehydrogenation to amide[56].

To further confirm the process of oxidative cyanation of thiols, in situ DRIFTS experiments with 4-*tert*-butylbenzyl mercaptan (**s6**) in flowing $O_2$ at 180 °C were performed under gas phase conditions (Fig. 7c). Initially, the characteristic peaks of C-H (tertiary butyl) and phenyl ring in 4-*tert*-butylbenzyl mercaptan be seen at 2964 $cm^{-1}$ and 1650–1300 $cm^{-1}$, respectively. As the reaction proceeded, a new peak at 1699 $cm^{-1}$ belonged to C = O group in benzaldehyde was observed. After a small amount of $NH_3$ was injected into the reaction, the absorption band at 3334 $cm^{-1}$, due to the breathing mode of N-H, appeared instantly, while the strength of the C = O stretch at 1699 $cm^{-1}$ gradually decreased. Meanwhile, an additional peak located at 2230 $cm^{-1}$ formed, which can be assigned to the characteristic absorption peak of benzonitrile, demonstrating the cyanation between benzaldehyde and ammonia. The oxidative cleavage and cyanation process of thiols confirmed by infra-red is consistent with our above-mentioned speculation.

To identify the type of active oxygen species generated over the catalyst, several control experiments were performed in the presence of PBQ, isopropanol, and $NaN_3$, which are commonly utilized as specific radical quenchers to singlet excited oxygen ($^1O_2$), hydroxyl radicals (•OH), and superoxide radicals ($O_2$•−)[57,58] (Fig. 7d). It was observed that the transformation of phenylmethanethiol **s1** to nitrile **1** was not affected in the presence of $NaN_3$ or isopropanol. On the contrary, with PBQ as the radical quenchers, transformation of **s1** was significantly inhibited. These results indicate that the oxidative cyanation reaction probably involves a superoxide radical-mediated oxidation step. Evidence for the generation of superoxide radical has been further confirmed by the electron paramagnetic resonance (EPR) spectroscopy using 5,5-dimethyl-1-pyrroline N-oxide (DMPO) as the capture agent[59,60] (Fig. 7e). In addition, when BHT was used as a radical

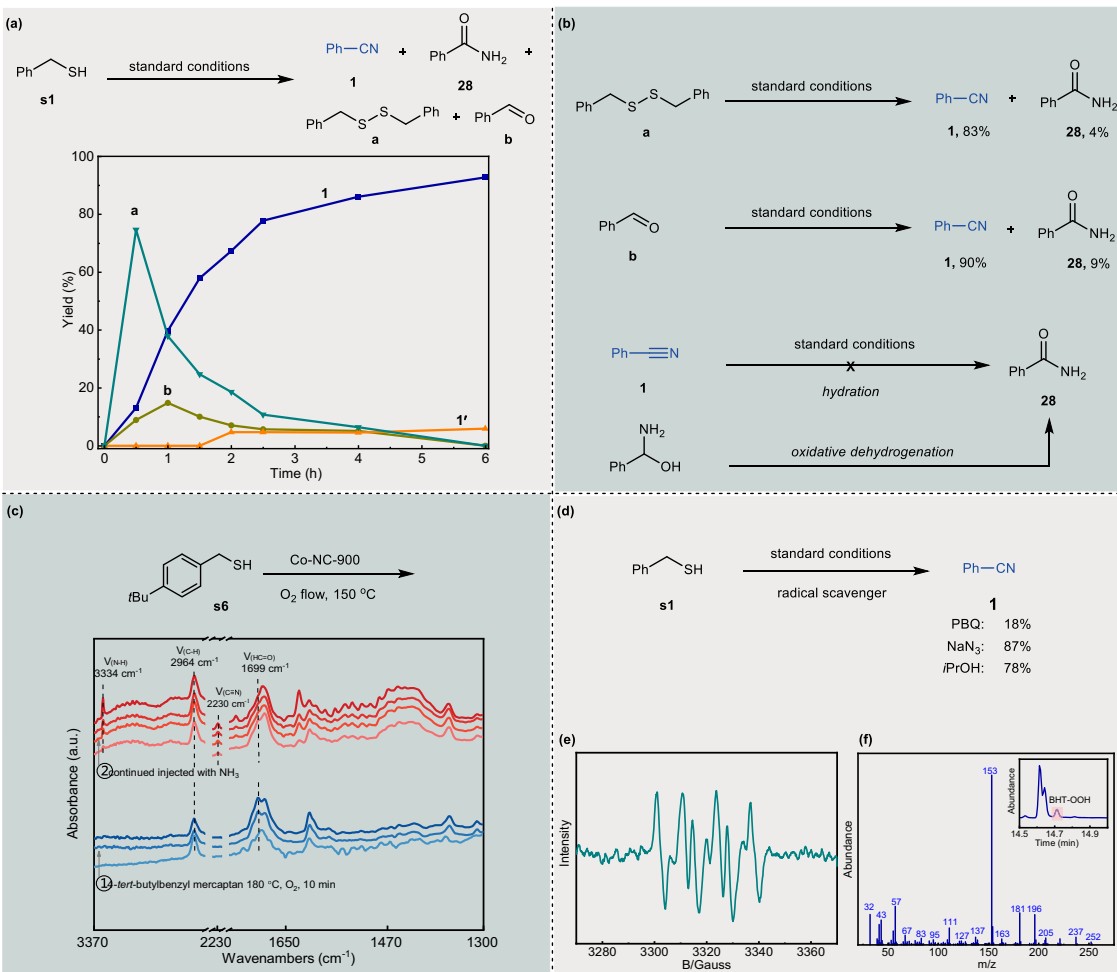

**Fig. 7 | Mechanistic investigations. a** Kinetic experiments. **b** Control experiments. **c** in situ DRIFTS spectra of the reaction of 4-*tert*-butylbenzyl mercaptan. **d** Effect of radical quenchers for the oxidative cleavage of phenylmethanethiol over Co-NC-900 catalyst with the addition of isopropanol, NaN₃ and PBQ. **e** EPR experiment performed with DMPO. **f** Radical inhibition experiments performed with BHT.

scavenger, an appreciable decrease in yield of target nitrile product **1** was observed. Indeed, the active superoxide radical O₂·• was captured by the BHT because BHT-OOH was detected by GC-MS (Fig. 7f). All of these results indicate that the generated O₂·• species is indispensable for this oxidative conversion.

Base on the above results and literature precedents, a putative mechanism for the oxidative cleavage and cyanation of organosulfur compounds is tentatively suggested (Fig. 8a). Initially, thiol **A** can be readily converted into thiyl radical under oxidation conditions, followed by radical coupling to give the disulfide **B**[18]. The dissolved molecular oxygen on the surface of the catalyst was reduced to produce the superoxide radical anion (O₂·•), which could then react with disulfide to afford dibenzylic radical anion **C** and hydroperoxyl radical (HOO•)[61]. The electron transfer between **C** and catalyst generates dibenzylic radical **D**[61]. Recombining the radical **D** with the hydroperoxyl radical results in the α-hydroperoxy disulfide **F**. At the same time, the **D** can also directly react with O₂ to form diperoxy radical **E**, then hydrogen abstraction by **E** from HOO• or thiol gives rise to **F**[62,63].

Next, the α-hydroperoxy disulfide **F** undergoes either an inter-or intramolecular oxygen transfer process to generate the α-hydroxy disulfide **G** and oxothiiranium ion **H**, respectively[16–18,64]. Both intermediates **G** and **H** were identified by high-resolution mass spectrum (HRMS) analyses (Supplementary Figs. 5, 6), and can rapidly decompose to form aldehyde **I**[16–18,64]. Nucleophilic attack of ammonia to aldehyde generates the hemiaminal intermediate **J**[53–55]. The hemiaminal intermediate **J** can be transformed via two pathways: the major

and minor one being the dehydration and oxidative dehydrogenation to aldimine **K** and amide **L**, respectively[53–55]. The aldimine **K** is unstable and readily undergoes the oxidative dehydrogenation to afford the nitrile **M** as the final product[53–55]. Additionally, the nitrile is inert to the hydrolysis process in the current catalytic system, which favors the high selectivity of C-S bond cyanation reaction.

Similar to the reaction pathways for thiol, the benzyl phenyl sulfide **N** can react with superoxide radical anion (O₂·•) to form the α-hydroperoxy sulfide intermediate **O**, which undergoes either an inter-or intramolecular oxygen transfer process to generate **P** and **Q**, respectively (Fig. 8b). The decompose of **P** and **Q** would lead to aldehyde **I** with the concurrent generation of the sulfenic acid **R** or thiophenol **S**[64]. The aldehyde can be efficiently converted into the nitrile **M** in the presence of O₂ an NH₃.

Meanwhile, both sulfenic acid **R** and thiophenol **S** can smoothly undergo amino-oxidation to give the corresponding sulfonamide **T**[65].

## Determination of active site

Based on the characterizations and mechanistic studies described above, we conducted a set of control experiments to identify the contribution of core-shell structured cobalt nanoparticles and atomically dispersed Co-Nx for the oxidative cleavage and cyanation reaction. First, the catalyst Co-NC-900 was leached with aqua regia to remove the metallic Co NPs, denoted as Co-NC-900-H⁺. Based on HRTEM images (Supplementary Fig. 4), no obvious nanoparticles were found in the acid-etched catalyst Co-NC-900-H⁺. Notably, the hollow-

**Fig. 8 | Proposed mechanism. a** Reaction pathways for thiol. **b** Reaction pathways for sulfide.

centered graphitic carbon layers were preserved in Co-NC-900-H⁺, further confirming the core-shell structure of the catalyst Co-NC-900. When the catalyst was changed to Co-NC-900-H⁺ for the reaction of **s1** under the standard conditions, a remarkable decrease in yield of **1** was observed, indicating that both Co NPs and Co-Nx are indispensable for achieving high catalytic activity (Fig. 9a). Subsequently, the reaction was carried out for 1 h using Co-NC-900-H⁺ as catalyst in the absence of aq. NH₃, affording intermediate benzaldehyde **a** in good yield. For comparison, the reaction of **s1** was conducted with the addition of KSCN, which is well known to poison the metal-Nx catalyst in oxygen reduction reaction[65,66], no appreciable decrease in the yield of ben-zaldehyde **a** was observed under otherwise identical conditions, implying that Co NPs are more active for the C-S bond oxidative cleavage to aldehyde intermediate (Fig. 9a). Finally, the intermediate aldehyde **a**, instead of thiol, was subjected to the standard reaction conditions, generating **1** in good yield within 1 h. Upon changing the catalyst to Co-NC-900-H⁺, deleterious effect on the yield **1** was observed under otherwise identical conditions. The yield of **1** was

further significantly decreased upon addition of KSCN, demonstrating that the Co-Nx exhibits better the activity towards the transformation of aldehyde intermediate to the nitrile (Fig. 9b). In addition, nano Co powder and Co(II)Pc were used as the analog of cobalt nanoparticles and Co-Nx in Co-NC-900, respectively, for the oxidative cleavage and cyanation reactions of organosulfur compounds. As expected, Co(II)Pc is more active in the oxidative cleavage step of the reaction process but with lower activity for further cyanation into the desired product, while nano Co powder was just in opposite position. Taken together, these results unambiguously corroborate that the remarkable effec-tiveness of the synergistic catalysis of Co nanoparticles and Co-Nx sites is responsible for excellent catalytic activity toward cleavage and cyanation reaction of the organosulfur compounds.

## Discussion
In this manuscript, we have successfully developed the first cyanide-free one step transformation of organosulfur compounds to nitriles by heterogeneous Co-NC catalysts via C-S bond cleavage. A wide variety

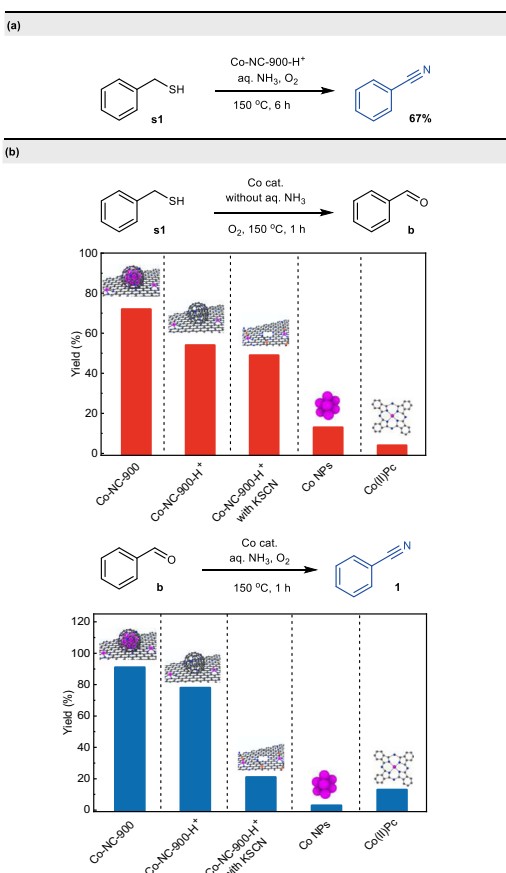

**Fig. 9 | Insight into the active sites of cobalt/N-doped carbon catalyst for the oxidative cleavage and cyanation reaction. a** Reaction of **s1** using acid-etched catalyst under standard conditions. **b** Comparison of catalytic performance of active sites for the key steps in the transformation.

of organosulfur compounds are viable in this reaction, enabling access a broad diversity of nitriles under cyanide-free conditions. Notably, the oxidative cleavage and amidation of organosulfur compounds to access amides can also be achieved through simply tuning the reaction conditions. The present transformation can be accomplished with $O_2$ as environmentally benign oxidant and ammonia as nitrogen source. In addition, characterization and mechanistic studies reveal that the remarkable effectiveness of the synergistic catalysis of Co nanoparticles and Co-Nx sites is crucial for achieving outstanding catalytic performance. This protocol features excellent functional group compatibility, facile scalability, cost-effective and recyclable catalyst, use of readily available starting materials, and a broad substrate scope. The present method not only expands the range of application of the sulfur-containing feedstocks, but also provides a new strategy for the divergent, late-stage cyanation of complex molecules. Further detailed mechanistic studies and applications based on this chemistry are in progress in our laboratory.

## Methods

### Materials
Cobalt (II) acetate tetrahydrate, aq. ammoniawere purchased from Sinopharm Chemical Reagent Co, Ltd. LUDOX®HS-40 colloidal silica was purchased from Sigma-Aldrich. 1,10-Phenanthroline monohydrate, Benzyl mercaptan, (4-Methylphenyl) methanethiol, (3-Methylphenyl)methanethiol, (2-Methylphenyl) methanethiol, 4-tert-Butylbenzyl mercaptan, 4-Methoxy-α-toluenethiol, and Thiophen-2-ylmethanethiol were purchased from Bide pharm Technology Co, Ltd. Phenylmethanethiol,

4-Chlorobenzyl mercaptan, 4-Bromobenzyl mercaptan, [4-(Trifluoromethyl)phenyl]methanethiol and 2,4-Dichlorobenzyl Mercaptan were purchased from Aladdin Bio-Chem Technology Co, Ltd. (4-Fluorophenyl)methanethiol, (3-Fluorophenyl)methanethiol, (2-Fluorophenyl)methanethiol, 2-Pyridinemethanethioland, 3-Pyridinemethanethiol were purchased from Macklin Bio-Chem Technology Co, Ltd. Benzyl Phenyl Sulfide, Benzyl(4-bromophenyl)sulfane, Benzyl Methyl Sulfide, Benzenesulfonylmethylbenzene, Phenylmethanesu-lfonamide, Benzyl sulfone, Dibenzyl Sulfoxide, Alpha-Toluenesulfonyl Chloride and 2-Methyl-2-butanol were purchased from Jiuding Bio-Chem Technology Co, Ltd. Benzyl methyl sulfone were purchased from J&K Scientific (Shanghai, China). All chemicals were used as received without further purification.

### Procedure for the preparation of catalysts
Co(OAc)$_2$·4H$_2$O (1.5 mmol) and 1,10-phenanthroline (3 mmol) were dissolved in 10 mL distilled water with vigorous magnetic stirring at room temperature for 10 min to obtain solution A. LUDOX®HS-40 colloidal silica (2.5 g) was dispersed in 40 mL of distilled water to obtain solution B. Next, solution A were dropwise added into solution B, and the whole reaction mixture was heated at 60 °C for 4 h. After evaporation of the solvent at 100 °C, the obtained Co(phen)$_2$(OAc)$_2$@SiO$_2$ was pyrolyzed at 900 °C (or 700, 800 °C) for 2 h under N$_2$ at a heating rate of 5 °C·min$^{-1}$. Finally, the black mesoporous alternatively treated with HF solution (20 wt%) to remove SiO$_2$ template, followed by washing with distilled water until the pH of the wash water was neutral. The Remaining material was filtrated, dried under vacuum at 100°C overnight, and denoted as Co-NC-X (X = pyrolysis temperature).

### General procedure for the synthesis of nitriles
In a typical experiment, the desired substrate, catalyst, 25-28 wt% aq. NH$_3$ and solvent were into a round-bottom flask (10 mL) with a magnetic bar. The autoclave was sealed, purged with O$_2$ to exclude the air three times, charged the O$_2$ pressure to 1.0 MPa. Subsequently, the autoclave was stirred at 150 °C for the required time. After the completion of the reaction, the vials were removed from the autoclave. Naphthalene as a standard was added, and the reaction product was diluted with t-amyl alcohol followed by centrifugation and then analyzed by GC and GC mass spectrometry (GC-MS).

### General procedure for the synthesis of amides
In a typical experiment, the desired substrate (0.25 mmol), Co-NC-900 (5.5 mol%), 3 Å molecular sieve (100 mg), and DMSO (1 mL) were into a round-bottom flask (10 mL) with a magnetic bar. The autoclave was sealed, purged with O$_2$ to exclude the air three times and then charged with O$_2$ (0.5 MPa) and NH$_3$ (0.5 MPa). Subsequently, the autoclave was stirred at 150 °C for the required time. The resulting solution was cooled in an ice bath and then H$_2$O (0.35 mL), 30% H$_2$O$_2$ (0.25 mL) and potassium carbonate (45 mg) were added, followed by allowing the reaction to warm up to room temperature. After 1 h, distilled water was removed under reduced pressure. Naphthalene as a standard was added, and the reaction product was diluted with MeOH followed by centrifugation and then analyzed by GC and GC mass spectrometry (GC-MS).

### Recovery and reuse of Co-NC-900
The recycling of catalyst experiments was carried out using phenylmethanethiol, as model substrate applying standard procedure under following reaction conditions: phenylmethanethiol (0.25 mmol), Co-NC-900 (5.5 mol%), 25-28 wt% aq. NH$_3$ (155 μL) and t-amyl alcohol (2 mL), 1.0 MPa O$_2$, 150 °C, 6 h. After completion of the reaction, in each run the catalyst was separated by centrifugation, washed with methanol and calcined at 900 °C under N$_2$ for 2 h. Then

the catalyst was used for the next run. Alternatively, after completion of the reaction, the reaction solution was carefully decanted and the fresh solvent, substrate and ammonia were added and the reaction was performed. Conversion and yield were determined by GC analysis using naphthalene as standard.

## Procedure for gram scale reaction

To a Teflon-fitted 100 mL autoclave, the desired substrate, catalyst, 25–28 wt% aq. $NH_3$ were transferred and then solvent was added. The autoclave was sealed, purged with $O_2$ to exclude the air three times, charged the $O_2$ pressure to 1.0 MPa. Subsequently, the autoclave was stirred at 150 °C for the required time. After the completion of the reaction, the vials were removed from the autoclave. Naphthalene as a standard was added, and the reaction product was diluted with tetrahydrofuran followed by centrifugation and then analyzed by GC and GC mass spectrometry (GC-MS).

## Characterization

Nuclear magnetic resonance spectra (NMR) were obtained on a Bruker Avance III 400 MHz instruments. $CDCl_3$ was used as solvent and tetramethylsilane (TMS) was used as the internal standard. Chemical shifts were reported in units (ppm) by assigning TMS resonance in the $^1H$ NMR spectrum as 0.00 ppm. The following abbreviations (or combinations thereof) were used to explain multiplicities: s = singlet, d = doublet, t = triplet, q = quartet, m = multiplet, b = broad. Coupling constants, J were reported in Hertz unit (Hz). Chemical shifts for $^{13}C$ NMR spectra were recorded in ppm from TMS using the central peak of $CDCl_3$ (77.0 ppm) as the internal standard. Chemical shifts are reported in ppm with the solvent resonance as the internal reference ($CDCl_3$ δ 77.16). HRMS were obtained with a Q-Tof analyzer. The X-ray power diffraction (XRPD) of samples were recorded on a Rigaku D/Max 2500PC diffractometer equipped with Cu Ka (λ = 1.5418 Å) at a scanning rate of 0.05°/s. The structure of catalysts was observed using a JEM-2100 transmission electron microscope (TEM) operated at an accelerating voltage of 120.0 kV. The high angle annular dark field scanning transmission electron microscopy (HAADF-STEM) images and EDS mapping were taken on an aberration-corrected JEM-ARM 200 F microscope.

## Data availability

All data needed to evaluate the conclusions in the paper are present in the paper and/or the Supplementary Information. Additional data are available from authors upon request.

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

## Acknowledgements
We thank the support of the Dalian Institute of Chemical Physics.

## Author contributions
W.D., H.W., and Hu.L. conceived and designed the experiments. Hu.L. and S.T. performed the experiments and analyzed the data. Ho.L.

designed and drew the structure diagram of the catalyst. S.G. participated in the discussions. W.D. and Hu.L. prepared the manuscript with feedback from all authors.

## Competing interests

The authors declare no competing interests
