## [Peer Review File · Nature Communications]

Oxidative Cleavage and Ammoxidation of Organosulfur compounds via Synergistic Co-Nx Sites and Co Nanoparticles CatalysisREVIEWER COMMENTS

Reviewer #1 (Remarks to the Author):

Dai and co-workers report the reaction of a variety of organosulfur compounds to benzonitriles using heterogeneous Co-NC catalysts. This novel transformation allows the synthesis of diverse benzonitriles in good yields and selectivity. Based on control experiments including kinetic studies the authors propose a convincing mechanistic proposal, too. Furthermore, recycling of the catalyst and applications on g-scale are shown. Clearly, the strength of this manuscript is the novelty of the shown methodology (catalytic reactions with sulfur compounds) and potential synthetic applications (interesting functional group tolerance, broad diversity). In general, I recommend publication of this nice piece of work after answering the following questions and revisions:

1. The authors should be aware of some drawbacks of the methodology at present, which are relatively high reaction temperature and more importantly relatively high loading of the heterogeneous materials and should communicate this also in their manuscript. What is the minimum amount of catalyst material necessary for the model reaction?
2. I assume the authors tested some substrates which did not work well or sluggishly. I suggest showing them in a Scheme in the SI.
3. With respect to 2, I am sure the authors have tested aliphatic thiols or thioethers in their methodology. They should comment on the outcome of the reactions. I assume sulfur oxidation took place, but no C-S cleavage.
4. It is clear that the cyanide free preparation of nitriles is interesting. However, using pure oxygen (especially at 150°C) I have some safety concerns. These should be checked. In addition, it would be interesting to perform an experiment using air as oxidant (which is more safe than pure oxygen).
5. In their detailed characterization the authors showed the HAADF-STEM image of the cobalt/graphene core-shell structure in Co-NC-900. Normally, such carbon shell can be expected to be oxidized under these conditions (O₂ and 150°C). Thus, how stable is the carbon shell, which should be easily seen from the TEM of the 3- or 5-fold recycled catalyst.

Reviewer #2 (Remarks to the Author):

The manuscript by Dai and coworkers describes a catalytic system based on Co nanoparticle/Co-N_x sites for oxidative desulfurization for the selective conversion of organosulfur compounds to corresponding nitrile and amide compounds. The reagents are aerobic oxygen and ammonia. This is an important

reaction and the catalytic system appears to be efficient, effective for a broad range of substrates and recyclable. Although the catalyst was synthesized earlier, a series of control experiments are carried out to gain insight on the mechanism. Although several spectroscopic studies have been carried out to identify the intermediates, I am not convinced that there are sufficient evidences to support the mechanism given in Fig. 8.

The other issue is the 'synergy' between the graphene-encapsulated Co nanoparticles and Co-Nx sites. Evidently both entities catalyze the reaction and the combination gives the best results. But this does not necessarily establish the 'synergistic' effect. Since this is claimed on the Title, more elaborate evidence is needed.

It is not obvious that Co nanoparticles and Co-Nx sites would catalyze in the same manner. What is observed is simply an average yield. The relative proportions of these two entities are known. It will be useful to discuss how mol% of the catalyst is calculated.

It is somewhat unusual to summarize the work under 'Discussion'. Results section includes both results and an integrated discussion. Perhaps it was meant as Conclusion.

In summary, this is an important work and should be published. But I do NOT think the originality and the depth of the work qualify it to be suitable for publication in a top-tier journal.

There are few instances of misspelling (minor point).

Response to the reviewers

Manuscript NCOMMS-22-44883

Title: "Oxidative C-S Bond Cleavage by Synergistic Co-Nx Sites and Co Nanoparticles Catalysis: An Efficient Synthesis of Nitriles and Amides"

Corresponding Author: Professor Wen Dai (Dalian Institute of Chemical Physics, Chinese Academy of Sciences)

Reviewer Comments:

Reviewer #1 (Remarks to the Author):

Dai and co-workers report the reaction of a variety of organosulfur compounds to benzonitriles using heterogeneous Co-NC catalysts. This novel transformation allows the synthesis of diverse benzonitriles in good yields and selectivity. Based on control experiments including kinetic studies the authors propose a convincing mechanistic proposal, too. Furthermore, recycling of the catalyst and applications on g-scale are shown. Clearly, the strength of this manuscript is the novelty of the shown methodology (catalytic reactions with sulfur compounds) and potential synthetic applications (interesting functional group tolerance, broad diversity). In general, I recommend publication of this nice piece of work after answering the following questions and revisions:

Response: We gratefully thank the reviewer for the insightful comments, which have helped us to improve the quality of the manuscript. Each suggested revision and comment, brought forward by reviewer, was considered. Below the comments of the reviewer are response point by point.

Question 1: The authors should be aware of some drawbacks of the methodology at present, which are relatively high reaction temperature and more importantly relatively high loading of the heterogeneous materials and should communicate this also in their manuscript. What is the minimum amount of catalyst material necessary for the model reaction?

Response: We are grateful for the suggestions. As mentioned in our paper, the model reaction was performed with a catalyst amount of 5.5 and 4 mol% can afford the desired nitrile **1** in 93 and 87% yields, respectively (Table R1, entries 1-2). Moderate yield of benzonitrile **1** (63%) was obtained when the catalyst loading was further decreased to 2.5 mol% (20 mg Co-NC-900) (entry 3). Gratifying, when the reaction time was elongated from 6 h to 12 h, good yield (81%) could also be achieved even with only 2.5 mol% catalyst (entry 4). Notably, lowering the temperature to 130 °C led to the **1** in acceptable yield (entry

5). Unfortunately, further lowering the temperature to 110 °C resulted in an obviously decrease in yield of **1**. These results suggest that in order to obtain good yield of **1** in the current catalyst system, no less than 2.5 mol% catalyst and 130 °C reaction temperature are indispensable.

Table R1. Investigation of reaction temperature and catalyst dosage in oxidative cleavage and cyanation of phenylmethanethiol^a

Entry	catalyst (mol%)	T (°C)	Time (h)	Yield of 1 (%) ^b
1	5.5	150	6	93
2	4.0	150	6	87
3	2.5	150	6	63
4	2.5	150	12	81
5	2.5	130	12	69
6	2.5	110	24	49

^aReaction conditions: substrates (0.25 mmol), NH₃-H₂O (155 μL), *t*-amyl alcohol (2 mL), 1.0 MPa O₂.

^bDetermined by GC analysis using biphenyl as internal standard and the products were confirmed by GC-MS.

Question 2: I assume the authors tested some substrates which did not work well or sluggishly. I suggest showing them in a Scheme in the SI.

Response: Thank you for the suggestions. Organosulfur compounds (**s82**, **s83**) were also evaluated, but no products were observed. As suggested by reviewer, these results have been added to the revised supporting information with yellow background highlighting.

Question 3: With respect to 2, I am sure the authors have tested aliphatic thiols or thioethers in their methodology. They should comment on the outcome of the reactions. I assume sulfur oxidation took place, but no C-S cleavage.

Response: The desulfurization of heavy oils is one of the challenges faced during refining to produce transportation fuels and petrochemicals. In order to further explore the application potential of this protocol, attempts were made to accomplish the oxidative transformation of inert aliphatic organic sulfur compounds such as aliphatic thiols (**s76-s79**), disulfides (**s80**) and sulfides (**s81**) which represent the main source of sulfur found

in heavy oils (Fig. R1). Although low yield (< 10%) of target nitriles or amides was obtained for the oxidative cleavage of C-S bond in these inert aliphatic organosulfur compounds, high conversion could be achieved to furnish the high-value-added sulfonamides in good yield. Sulfonamides are more polar than other hydrocarbons in crude oils, and thus can be simply removed from oil phase by extractive desulfurization. These results demonstrate that the current catalytic oxidation system can provide an alternative strategy for the oxidative desulfurization of heavy oil.

Fig. R1. Desulfurization of sulfur-containing compounds in heavy oil. Reaction conditions: substrate (0.25 mmol), Co-NC-900 (11 mol%), 25-28 wt% aq. NH_3 (155 μL), *t*-amyl alcohol (2 mL), 1.0 MPa O_2 , 150 $^{\circ}C$, 48 h. ^asubstrate (0.5 mmol), Co-NC-900 (5.5 mol%), 25-28 wt% aq. NH_3 (170 μL), acetonitrile (0.5 mL), 150 $^{\circ}C$, 16 h. ^bsubstrate (0.5 mmol), Co-NC-900 (5.5 mol%), 25-28 wt% aq. NH_3 (170 μL), acetonitrile (0.5 mL), 170 $^{\circ}C$, 48 h.

Question 4: It is clear that the cyanide free preparation of nitriles is interesting. However, using pure oxygen (especially at 150 $^{\circ}C$) I have some safety concerns. These should be checked. In addition, it would be interesting to perform an experiment using air as oxidant (which is more safe than pure oxygen).

Response: Thank you for the suggestions. As suggested by reviewer, we performed the reactions of oxidative cleavage and cyanation of different types of organosulfur compounds using air as oxidant. As depicted in Fig R2, a wide variety of thiols (**s1**, **s3**, **s11**, **s17**), sulfides (**s20**, **s33**, **s34**), sulfoxides (**s53**), sulfones (**s56**), and sulfonamides (**s60**) can be efficiently transformed into the corresponding nitriles in high to excellent yields under an air atmosphere.

Fig. R2 Cyanation of organosulfur compounds using air as oxidant. Yield was determined by GC analysis using biphenyl as internal standard and the product was confirmed by GC-MS. Reaction conditions: substrate (0.25 mmol), Co-NC-900 (5.5 mol%), 25-28 wt% aq. NH_3 (155 μL), *t*-amyl alcohol (2 mL), 1.0 MPa air, 150 $^\circ\text{C}$, 6 h. ^aCo-NC-900 (11 mol%), 24 h. ^bCo-NC-900 (11 mol%), 25-28 wt% aq. NH_3 (233 μL), 24 h.

Question 5: In their detailed characterization the authors showed the HAADF-STEM image of the cobalt/graphene core-shell structure in Co-NC-900. Normally, such carbon shell can be expected to be oxidized under these conditions (O_2 and 150 $^\circ\text{C}$). Thus, how stable is the carbon shell, which should be easily seen from the TEM of the 3- or 5-fold recycled catalyst.

Response 5: We are grateful for the suggestion. As suggested by reviewer, the 3- and 5-fold recycled catalyst has been characterized by High-resolution TEM (HRTEM). As illustrated in the following figures (Fig. R3), no obvious change of the lattice fringes of carbon shell were observed in comparison to the fresh catalyst. These results unambiguously demonstrate the carbon shell possess excellent stability.

Figure R3. HRTEM images of fresh (a), 3-fold reused (b) and 5-fold reused (c) Co-NC-900.

Reviewer #2 (Remarks to the Author):

The manuscript by Dai and coworkers describes a catalytic system based on Co nanoparticle/Co-Nx sites for oxidative desulfurization for the selective conversion of organosulfur compounds to corresponding nitrile and amide compounds. The reagents are aerobic oxygen and ammonia. This is an important reaction and the catalytic system appears to be efficient, effective for a broad range of substrates and recyclable. Although the catalyst was synthesized earlier, a series of control experiments are carried out to gain insight on the mechanism. Although several spectroscopic studies have been carried out to identify the intermediates, I am not convinced that there are sufficient evidences to support the mechanism given in Fig. 8.

The other issue is the 'synergy' between the graphene-encapsulated Co nanoparticles and Co-Nx sites. Evidently both entities catalyze the reaction and the combination gives the best results. But this does not necessarily establish the 'synergistic' effect. Since this is claimed on the Title, more elaborate evidence is needed.

It is not obvious that Co nanoparticles and Co-Nx sites would catalyze in the same manner. What is observed is simply an average yield. The relative proportions of these two entities are known. It will be useful to discuss how mol% of the catalyst is calculated.

It is somewhat unusual to summarize the work under 'Discussion'. Results section includes both results and an integrated discussion. Perhaps it was meant as Conclusion.

In summary, this is an important work and should be published. But I do NOT think the originality and the depth of the work qualify it to be suitable for publication in a top-tier journal.

There are few instances of misspelling (minor point).

Response: The authors would like to thank the reviewer for recognizing the importance of this work and providing valuable suggestions which have significantly improved our manuscript. With the reviewer's suggestions in mind, we have carried out more experiments to support the reaction mechanism (identification of reactive intermediates by HRMS analysis) (See Fig R1-2) and the 'synergy' between the graphene-encapsulated Co nanoparticles and Co-Nx sites (See Fig R5). In addition, to further highlight the potential application and importance of our protocol, cyanation reactions of different types of organosulfur compounds by replacing pure oxygen with cheaper, safer and more easily handled air as oxidant (See Fig R6) and oxidative transformation of aliphatic thiols and disulfides which represent the main source of sulfur found in heavy oils (See Fig R7) have been added in the revised manuscript accordingly, which significantly strengthen the paper.

Here we would emphasize that the most notable merits of our manuscript include:

First and most importantly, this is one of the only catalytic methods that enables direct synthesis nitriles or amides via oxidative cleavage of C-S bonds in organosulfur compounds. Second, the present chemistry represents a significant advancement in developing efficient and sustainable heterogeneous catalysts for the late-stage cyanation of sulfur-containing complex molecules (pharmaceuticals and natural products) and oxidative desulfurization of heavy oil. Third, this protocol features excellent functional group compatibility, facile scalability, cost-effective and recyclable catalyst, use of readily available starting materials, and a broad substrate scope.

Question 1: Although the catalyst was synthesized earlier, a series of control experiments are carried out to gain insight on the mechanism. Although several spectroscopic studies have been carried out to identify the intermediates, I am not convinced that there are sufficient evidences to support the mechanism given in Fig. 8.

Response: We are grateful for the suggestion. To gain a better understanding of the mechanism given in Fig. 8, the reaction of phenylmethanethiol was conducted in the absence of NH_3 , and α -hydroxy disulfide **G** and oxothiiranium ion **H** were detected by high-resolution mass spectrum (HRMS) (Fig. R1-2), which are important reactive intermediates for the oxidative cleavage of organosulfur compounds to aldehydes according to previous reports. (*J. Am. Chem. Soc.* **122**, 1834-1835 (2000); *J. Am. Chem. Soc.* **123**, 4966-4973 (2001); *Org. Lett.* **22**, 4395-4399 (2020)).

Fig. R1 HRMS (ESI) m/z $[M+NH_4]^+$ calculated for $C_{14}H_{18}NO_2S_2$ 296.0779, found 296.0795.

HRMS (ESI) m/z $[M+Na]^+$ calculated for $C_{14}H_{14}NaO_2S_2$ 301.0333, found 301.0349.

Fig. R2 (a) HRMS (ESI) m/z $[M+NH_4]^+$ calculated for $C_{14}H_{18}NO_4S_2$ 328.0677, found 328.0688.

(b) HRMS (ESI) m/z $[M+Na]^+$ calculated for $C_{14}H_{14}NaO_4S_2$ 333.0231, found 333.0249.

As mentioned in our paper, a series of kinetic experiments, control experiments, and *in-situ* Fourier transform infrared (FT-IR) spectra studies (Fig. R3a-c) revealed that disulfide **B** and aldehyde **I** are clearly the intermediates of the oxidative cleavage of thiol. **Moreover**, the formation of α -hydroxy disulfide **G** and oxythio iridium ion **H** was detected by

HRMS (newly added) (Fig. R1-2). In addition, radical inhibition experiments and EPR experiments (Fig. R3d-e) indicated that the generated $O_2\cdot$ -species was indispensable for this oxidative conversion. Therefore, a rational reaction mechanism of the oxidative cleavage and amoxidation of C-S bond was proposed based on these mechanistic experiments and previous studies (*J. Am. Chem. Soc.* **122**, 1834-1835 (2000); *J. Am. Chem. Soc.* **123**, 4966-4973 (2001); *Org. Lett.* **22**, 4395-4399 (2020); *J. Am. Chem. Soc.* **132**, 16299-16301 (2010); *Nat Commun* **9**, 933 (2018); *Angew. Chem. Int. Ed.* **57**, 14240-14244 (2018)), in Fig. 8 (Fig. R4).

Fig. R3 Mechanistic investigations. a Kinetic experiments. b Control experiments. c *in situ* DRIFTS spectra of the reaction of 4-*tert*-Butylbenzyl mercaptan. d Effect of radical quenchers for the oxidative cleavage of phenylmethanethiol over Co-NC-900 catalyst with the addition of isopropanol, NaN₃ and PBQ. e EPR experiment performed with DMPO. f Radical inhibition experiments performed with BHT.

Figure R4 Proposed mechanism of the oxidative cleavage and ammoxidation of thiol over the Co-NC-900 catalyst.

Question 2: The other issue is the 'synergy' between the graphene-encapsulated Co nanoparticles and Co-N_x sites. Evidently both entities catalyze the reaction and the combination gives the best results. But this does not necessarily establish the 'synergistic' effect. Since this is claimed on the Title, more elaborate evidence is needed.

Response: We are grateful for the suggestion. As suggested by reviewer, nano Co powder and Co(II)Pc were used as the analog of cobalt nanoparticles and Co-N_x in Co-NC-900, respectively, for the oxidative cleavage and cyanation reactions of organosulfur compounds. As expected, Co(II)Pc is more active in the oxidative cleavage step of the reaction process but with lower activity for further cyanation into the desired product, while nano Co powder was just in opposite position (Fig. R5). These results further demonstrate that the remarkable effect of the synergistic catalysis of cobalt nanoparticles and Co-N_x sites for the superior catalytic activity in the cracking and cyanation reactions of organosulfur compounds.

Fig. R5 Comparison of catalytic performance of active sites for the key steps in the transformation.

Question 3: It is not obvious that Co nanoparticles and Co-N_x sites would catalyze in the same manner. What is observed is simply an average yield. The relative proportions of these two entities are known. It will be useful to discuss how mol% of the catalyst is calculated.

Response: We are grateful for the suggestion. ICP-AES (inductively coupled plasma atomic emission spectroscopy) analysis revealed Co loadings of 2.04 and 1.15 wt % for the Co-NC-900 and Co-NC-900-H⁺, respectively (Table R2). Taking all characterization results into account, we can conclude that the as-prepared catalyst Co-NC-900 comprises core-shell structured nanoparticles with metallic Co as the core and layers of graphitic carbon as the shell and coordinated Co-N_x sites as well. HRTEM images of the acid-etched catalyst Co-NC-900-H⁺ show that no obvious nanoparticles were found. Based on these results, we hypothesize that the content of Co nanoparticles in Co-NC-900 is 0.89 wt%.

Table R2. ICP-OES analysis of Co-NC-900.

	Co content (wt%)	Catalyst (mol%)
Co-NC-900	2.04	5.5
Co-N _x (Co-NC-900-H ⁺)	1.15	3.13
Co NPs	0.89	2.37

The specific calculation methods of the catalyst (mol%) are as follows:

$$\text{Catalyst (mol\%)} = \frac{m \times c}{M \times n}$$

m = mass of catalyst

c = mass percentage of Co in catalyst

M = molecular weight of Co

n = molar quantity of substrate

Question 4: It is somewhat unusual to summarize the work under 'Discussion'. Results section includes both results and an integrated discussion. Perhaps it was meant as Conclusion.

Response: We have followed the reviewer's suggestions. The summary of the work has been corrected to 'Conclusion' in the revised manuscript with yellow background highlighting.

Question 5: There are few instances of misspelling (minor point).

Response: Thanks for your careful checks. We apologize for our careless mistakes. In our resubmitted manuscript, the misspelling has been revised.

Complementary applications:

1) Air is cheaper, safer, and more easily handled in comparison to pure oxygen. we performed the reactions of oxidative cleavage and cyanation of different types of organosulfur compounds using air as oxidant. As depicted in Fig R6, a wide variety of thiols (**s1**, **s3**, **s11**, **s17**), sulfides (**s20**, **s33**, **s34**), sulfoxides (**s53**), sulfones (**s56**), and sulfonamides (**s60**) can be efficiently transformed into the corresponding nitriles in high to excellent yields under an air atmosphere.

Fig. R6 Cyanation of organosulfur compounds using air as oxidant. Yield was determined by GC analysis using biphenyl as internal standard and the product was confirmed by GC-MS. Reaction conditions: substrate (0.25 mmol), Co-NC-900 (5.5 mol%), 25-28 wt% aq. NH₃ (155 μ L), *t*-amyl alcohol (2 mL), 1.0 MPa air, 150 $^{\circ}$ C, 6 h. ^aCo-NC-900 (11 mol%), 24 h. ^bCo-NC-900 (11 mol%), 25-28 wt% aq. NH₃ (233 μ L), 24 h.

2) The desulfurization of heavy oils is one of the challenges faced during refining to

produce transportation fuels and petrochemicals. In order to further explore the application potential of this protocol, attempts were made to accomplish the oxidative transformation of inert aliphatic organic sulfur compounds such as aliphatic thiols (**s76-s79**), disulfides (**s80**) and sulfides (**s81**) which represent the main source of sulfur found in heavy oils (Fig. R7). Although low yield (< 10%) of target nitriles or amides was obtained for the oxidative cleavage of C-S bond in these inert aliphatic organosulfur compounds, high conversion could be achieved to furnish the high-value-added sulfonamides in good yield. Sulfonamides are more polar than other hydrocarbons in crude oils, and thus can be simply removed from oil phase by extractive desulfurization. These results demonstrate that the current catalytic oxidation system can provide an alternative strategy for the oxidative desulfurization of heavy oil.

Fig. R7. Desulfurization of sulfur-containing compounds in heavy oil. Reaction conditions: substrate (0.25 mmol), Co-NC-900 (11 mol%), 25-28 wt% aq. NH_3 (155 μL), *t*-amyl alcohol (2 mL), 1.0 MPa O_2 , 150 $^\circ\text{C}$, 48 h. ^asubstrate (0.5 mmol), Co-NC-900 (5.5 mol%), 25-28 wt% aq. NH_3 (170 μL), acetonitrile (0.5 mL), 150 $^\circ\text{C}$, 16 h. ^bsubstrate (0.5 mmol), Co-NC-900 (5.5 mol%), 25-28 wt% aq. NH_3 (170 μL), acetonitrile (0.5 mL), 170 $^\circ\text{C}$, 48 h.

REVIEWERS' COMMENTS

Reviewer #1 (Remarks to the Author):

The authors have convincingly improved the manuscript and answered most of the questions of all the referees. The performed additional experiments improved the quality of the work and I advise publication in its present form.

Response to referees

Manuscript NCOMMS-22-44883A

Title: "Oxidative C-S Bond Cleavage by Synergistic Co-Nx Sites and Co Nanoparticles Catalysis: An Efficient Synthesis of Nitriles and Amides"

Corresponding Author: Professor Wen Dai (Dalian Institute of Chemical Physics, Chinese Academy of Sciences)

Response to reviewer

Reviewer Comments:

Reviewer #1 (Remarks to the Author):

The authors have convincingly improved the manuscript and answered most of the questions of all the referees. The performed additional experiments improved the quality of the work and I advise publication in its present form.

Response: We gratefully thank you for very positive comments on our work and suggesting publishing in its present form.